# B Cells in Primary Membranous Nephropathy: Escape from Immune Tolerance and Implications for Patient Management

**DOI:** 10.3390/ijms222413560

**Published:** 2021-12-17

**Authors:** Benjamin Y. F. So, Desmond Y. H. Yap, Tak Mao Chan

**Affiliations:** Department of Medicine, Division of Nephrology, Queen Mary Hospital, The University of Hong Kong, Hong Kong, China; soyufaibenjamin@gmail.com (B.Y.F.S.); dftmchan@hku.hk (T.M.C.)

**Keywords:** B cells, primary membranous nephropathy, immune tolerance

## Abstract

Membranous nephropathy (MN) is an important cause of nephrotic syndrome and chronic kidney disease (CKD) in adults. The pathogenic significance of B cells in MN is increasingly recognized, especially following the discovery of various autoantibodies that target specific podocytic antigens and the promising treatment responses seen with B cell depleting therapies. The presence of autoreactive B cells and autoantibodies that bind to antigens on podocyte surfaces are characteristic features of MN, and are the result of breaches in central and peripheral tolerance of B lymphocytes. These perturbations in B cell tolerance include altered B lymphocyte subsets, dysregulation of genes that govern immunoglobulin production, aberrant somatic hypermutation and co-stimulatory signalling, abnormal expression of B cell-related cytokines, and increased B cell infiltrates and organized tertiary lymphoid structures within the kidneys. An understanding of the role of B cell tolerance and homeostasis may have important implications for patient management in MN, as conventional immunosuppressive treatments and novel B cell-targeted therapies show distinct effects on proliferation, differentiation and reconstitution in different B cell subsets. Circulating B lymphocytes and related cytokines may serve as potential biomarkers for treatment selection, monitoring of therapeutic response and prediction of disease relapse. These recent advances in the understanding of B cell tolerance in MN have provided greater insight into its immunopathogenesis and potential novel strategies for disease monitoring and treatment.

## 1. Introduction

Membranous nephropathy (MN) is the most common cause of idiopathic nephrotic syndrome in adults [1]. Globally, the incidence rate of MN is up to 12 per million adults per year, although there is significant regional variation [1,2,3,4,5,6,7,8,9,10,11,12,13,14,15,16,17,18]. The incidence may be underestimated because only a small proportion of patients with MN progress to end-stage kidney disease (ESKD). Adults over the age of 50 are more commonly affected [19,20,21]. MN is defined by its characteristic histopathological features, which include thickened glomerular basement membrane and subepithelial spikes on the outer surface of the capillary wall. Immunofluorescence typically reveals granular deposits of IgG4 along the outer surface of the capillary walls, though IgG1 and IgG3 may be seen as well especially in earlier disease; complement components especially C3 may also be present. Electron microscopy confirms the diagnosis with identification of subepithelial electron-dense deposits, in conjunction with other pathological findings including glomerular basement membrane thickening and effacement of podocytes [22]. Clinically, MN is characterized by proteinuria, which can be of variable severity but often presents as full-blown nephrotic syndrome. The natural history of MN is variable: although around one-third of cases remit spontaneously, a significant number of patients manifest persistently heavy proteinuria [23,24,25]. A subset of patients who remain persistently nephrotic in spite of treatment may develop progressive chronic kidney disease (CKD) and eventually reach ESKD.

Traditionally, MN has been classified as primary or secondary. Common secondary causes of MN include malignancy, infections (such as hepatitis B and C), other systemic autoimmune diseases (such as systemic lupus erythematosus and sarcoidosis), and drugs; the label of “primary MN” was reserved for idiopathic cases for which extensive workup has not revealed an underlying cause [26]. The identification of pathognomonic antibodies in ‘idiopathic’ MN has cast doubt on the appropriateness of this terminology. Also, recent data suggest significant overlap between the pathophysiology underlying both the primary and secondary forms of MN, and an arbitrary distinction between primary and secondary forms may be overly simplistic [27,28].

A growing body of evidence suggests that MN is a kidney-specific autoimmune disease, arising from a loss of normal immune tolerance to podocyte antigens, with formation of disease-causing antibodies that result in a pathognomonic pattern of injury in glomeruli [25,29,30,31]. The role of B cells in the immunopathogenesis of primary MN is therefore increasingly recognized, and therapeutic depletion of B cells has gained prominence as an effective means of treating MN. This review will explore the mechanisms by which normal B cell tolerance is breached in MN and highlight the implications for clinical management.

## 2. Reconceptualizing Membranous Nephropathy as a B Cell Disorder

The identification of disease-specific podocyte antigens and associated pathogenic antibodies, as well as the success of B cell-targeted treatments, especially anti-CD20 monoclonal antibodies such as rituximab, suggest that B cells play a prominent role in MN. Historically, disease-causing autoantibodies directed against megalin in murine podocytes were isolated in the archetypal murine Heymann nephritis model of MN [32,33]. In the past two decades, the phospholipase-A_2_ receptor (PLA_2_R) has been found to be the culprit podocyte antigen in about 70–80% cases labeled as primary MN in human. Circulating anti-PLA_2_R antibodies, which are IgG4 antibodies targeted at different epitopes on PLA_2_R, are found in a majority of such cases [29,34,35,36,37,38]. Thrombospondin type-1 domain-containing 7A (THSD7A) is another target antigen implicated in about 5% of cases, and is likewise associated with circulating antibodies [39,40]. Other novel antigens identified include neural epidermal growth factor-1 (NELL1), exostosin 1/exostosin 2 (EXT1/EXT2), semaphorin 3B (Sema3B), and protocadherin-7 (PCDH7); circulating antibodies to some of these antigens have been isolated from the sera of patients with MN [41,42,43,44,45,46]. A growing number of studies suggest that these antibodies may be directly pathogenic in MN, and carry diagnostic, prognostic and therapeutic significance. Meanwhile, randomized controlled trials have confirmed the efficacy of anti-CD20 monoclonal antibodies, particularly rituximab, in MN, especially in but not limited to PLA_2_R-associated disease [47,48,49,50,51,52,53,54,55,56,57]. Smaller case series suggest similar or even superior efficacy with the use of other monoclonal antibodies targeted against B cell antigen(s) or B cell activating factor, such as obinutuzumab, ofatumumab and belimumab [58,59,60,61]. Taken together, these findings suggest that autoantibodies generated by autoreactive B cells are a key driver in the immunopathogenesis of MN. Although there is mounting data to support that breaches in normal B cell tolerance occur in MN and the downstream effects lead to glomerular injury, the exact underlying mechanisms remain elusive. The following discussion will overview the normal mechanisms for central and peripheral B cell tolerance, and summarize the current evidence for dysregulation of normal B cell tolerance in the context of MN.

## 3. Mechanisms of Central and Peripheral B Cell Tolerance

B lymphocytes are formed in the bone marrow and arise from progenitor cells known as pro-B cells that are committed to the B cell lineage. A key process of B cell development in the bone marrow is rearrangement of the immunoglobulin heavy chain (IgH) and light chain (IgL) to generate a diverse repertoire of immunoglobulin (Ig) specificities [62]. Immature B cells expressing innocuous B cell receptors (BCR) with high phosphoinositide-3-kinase (PI3K) activity are then positively selected for by tonic BCR signaling. However, as V(D)J recombination is a random process, the Ig repertoire generated cannot be predicted. A significant proportion of antigens bound by the BCR expressed on naïve B cells are necessarily self-antigens within the marrow microenvironment. Thus, negative selection is needed in the bone marrow to ligate BCR that bind self-antigens. The regulatory mechanisms in the bone marrow to reduce self-reactivity are collectively known as central tolerance [63]. If an immature B cell encounters a self-antigen at high concentrations in the bone marrow environment, the BCR cross-links, resulting in one of two fates. First, the B cell may undergo apoptosis, a process known as clonal deletion [64,65]. Second, the B cell may undergo another round of IgL gene recombination to generate a new, innocuous IgL specificity; this is known as receptor editing [66,67,68,69,70]. Alternatively, if the immature B cell encounters a self-antigen at low concentrations, the cell may simply become functionally unresponsive, or anergic, due to downregulation of the BCR and its downstream signaling pathways. The cell’s lifespan is then significantly shortened and apoptosis occurs within days [71,72]. These mechansims of negative selection of autoreactive B cells are summarized in Figure 1. The three main mechanisms of clonal deletion, receptor editing and anergy involved in negative selection are limited in that only self-antigens that are expressed in the bone marrow are tolerized. Thus, immature B cells that leave the bone marrow may yet carry autoreactive BCR specificities, and further tolerance checkpoints are needed in the periphery [63,73].

Immature B cells that exit the bone marrow home to the spleen, and differentiate into transitional B cells. These transitional B cells undergo further maturation and differentiation in the spleen into follicular B (FOB) cells or marginal zone B (MZB) cells [74]. This process is triggered by BCR-related pathways of signal transduction, but also by other critical cytokine pathways, such as the BAFF (B cell activating factor belonging to the tumor necrosis factor family) system and the NOTCH2 (neurogenic locus notch homolog protein 2) signaling pathway [75,76,77]. Under tonic, or intermediate-to-strong BCR-mediated signals and BAFF, and expression of transcription factors such as NOTCH2 and BTK (Bruton’s tyrosine kinase), transitional B cells can evolve into FOB cells, before being recirculated and occupying secondary lymphoid organs (SLOs) such as lymph nodes and mucosa-associated lymphoid tissues [78]. The most important role of FOB cells is to interact with follicular helper T cells (T_FH_ cells) in SLOs to stimulate further B cell differentiation and proliferation. B cells that are stimulated by an appropriate antigen receive cognate help from T_FH_ cells at the boundary of the lymphoid follicle adjacent to the T cell zone [79]. Such activated B cells may either develop into extrafollicular plasmablasts or early memory B cells, or may enter the follicles to form germinal centres with T_FH_ cells [80]. In germinal centres, B cells undergo somatic hypermutation (SHM) and/or class switch recombination (CSR) to generate a diverse repertoire of high-affinity antibodies. SHM refers to a process of stepwise incorporation of single-nucleotide substitutions into the V gene, catalyzed by the enzyme activation-induced deaminase (AID), resulting in an expanded diversity of antibodies [81]. CSR is another process catalyzed by the enzyme AID, whereby B cell activation results in the switching of the IgM isoform to other Ig isotypes, including IgG1/2/3/4, IgA1/2, IgD, and IgE, to form different antigen avidities and immune responses [82,83]. CSR takes place in the germinal centre and requires close cooperation between the FOB and T_FH_ cells via binding of the B cell surface protein CD40 to its ligand CD40L, which is expressed on T_FH_ cells [84]. Through these mechanisms, germinal centre B cells with increased affinity for the target antigen are selected and preferentially expanded, and differentiate into memory B cells and antibody-producing plasma cells [85]. Meanwhile, under weak BCR signaling, NOTCH2 signaling and NF-κB signals, transitional cells can also evolve into MZB cells in the splenic marginal zone [75]. MZB cells function as antigen-presenting cells to activate T_FH_ cells and can also differentiate into short-lived plasmablasts in response to blood antigens and form a large number of IgM. Some cells, such as dendritic cells, macrophages and NK/T cells can also trigger CSR in MZB cells [86]. Delta-like 1 (DL-1), the ligand of NOTCH2 on endothelial cells, is a key activator of MZB cells, and MZB migration to the follicles is also promoted by sphingosine-1-phosphate receptor (S1PR1) and CXC chemokine ligand 13 (CXCL13) [77,87].

As alluded to above, B cell selection and survival is regulated by several key cytokine systems, including the BAFF and APRIL (a proliferation-inducing ligand) [86,88,89], both belonging to the tumor necrosis factor (TNF) superfamily. BAFF, also known as BlyS (B lymphocyte stimulator) or TALL-1 (TNF and apoptosis ligand-related leukocyte-expressed ligand 1), is a cytokine expressed by monocytes, macrophages, dendritic cells, bone marrow stromal cells and T cells. Both BAFF and APRIL interact with TACI (transmembrane activator and cyclophilin ligand interactor) and BCMA (B cell maturation antigen), and BAFF additionally binds to the BAFF receptor (BAFF-R). BAFF-R is expressed by B cells starting from the time they evolve into transitional B cells and leave the bone marrow. TACI is expressed in activated B cells, marginal zone B cells, switched B cells and plasma cells, whereas BCMA is upregulated in activated B cells, and long-lived plasma cells requires APRIL or BAFF signaling for survival [90]. Through these receptors, BAFF serves as an important pro-survival signal for B cells through activation of the non-canonical NF-κB2 pathway and the PI3K pathway; absence of BAFF leads to an almost complete arrest of B cell development [89]. Logically, elevated serum BAFF levels have been associated with autoimmunity in B cell disorders such as systemic lupus erythematosus (SLE) [91]. As bone marrow B cells have very low levels of BAFF-R expression, BAFF expression levels have little impact on clonal deletion or receptor editing in the bone marrow. Therefore, perturbations in BAFF likely affect peripheral more than central tolerance of B cells [89]. Of note, autoreactive B cells are typically rendered anergic after repeated stimulation by self-antigen and are particularly dependent on high levels of BAFF for survival; the corollary is that they are more likely to be outcompeted by other, less BAFF-dependent naïve B cells for survival in lymphoid follicles [92]. Finally, TACI can serve as a decoy receptor but can also trigger CSR in germinal centres [93].

In addition to those that escape central tolerance mechanisms in the bone marrow, a significant population of autoreactive B cells is also generated due to the efficient process of SHM. Peripheral tolerance mechanisms are therefore necessary to prevent development of autoimmunity. These mechanisms include clonal deletion in the periphery, anergy, immunomodulation by regulatory T (Treg) or regulatory B (Breg) cells, and ignorance by cognate T cells. Escape from these immune tolerance mechanisms may occur in the setting of elevated BAFF levels, or augmented T cell help of anergic B cells [94]. Regulatory B cells that secrete regulatory cytokines, such as IL-10, transforming growth factor-β (TGF-β), and FoxP3, may also play a role in attenuating maladaptive immune and inflammatory responses, and their dysfunction is associated with development of autoimmunity [95,96].

Clearly, failure of any parts of the tightly orchestrated process of B cell development and regulation can result in generation of autoreactive B cells that contribute to the pathogenesis of MN. These defects are well described in conditions such as SLE, but are also increasingly recognized in the context of MN and will be explored in detail in the following sections. Modulation of these mechanisms of tolerance in B cells also has important therapeutic implications in MN.

## 4. Perturbations in Circulating B Cell Repertoire, Tolerance and Regulation in Membranous Nephropathy

Accumulating evidence suggests that the B cell repertoire is disturbed in MN. Although the absolute number of circulating B cells in the peripheral blood is not significantly altered in MN, the B cell population may be polarized towards naïve B cells, with decreased switched and non-switched memory B cells [97,98]. The circulating plasma cell population is also abnormally expanded in MN [99,100]. These findings are similar to those found in other autoimmune diseases including SLE, suggesting alterations in B cell tolerance checkpoints [101]. The reduction in circulating memory B cells in spite of autoimmunity could suggest either infiltration into end-organs such as the kidney, or by preferential differentiation of self-reactive B cells into plasmablasts rather than memory B cells. Infiltrating B cell subsets will be explored in greater depth in a later section, while PLA_2_R-specific IgG-producing plasmablasts were identified in the circulation of MN patients in a recent study [99].

A transcriptomic analysis demonstrated that RNA transcripts of multiple genes governing IgH status were differentially expressed in peripheral blood mononuclear cells (PBMCs) of patients with MN, as compared to healthy controls. Specifically, the frequency of IGHM, IGHD, and IGHE—genes that govern expression of μ, δ, and ε heavy chains on IgM, IgD and IgE respectively—were higher in MN patients, whereas the frequency of IGHA and IGHG4—genes that govern expression of α and γ_4_ heavy chains on IgA and IgG4 respectively—were lower in MN patients. In addition, usage pattern of IGHV genes coding for the immunoglobulin heavy-chain variable region were skewed, with a lower frequency of IGHV3 and comparatively higher frequency of IGHV4 [102]. These findings mirror the results of BCR sequencing studies in SLE, as lupus patients typically show enrichment in usage of the IGHV4 gene family [103,104]. Extrapolating from previous studies in SLE patients, such a usage pattern is typically associated with autoreactivity, although this can also be modified by disease stage and immunomodulatory treatment. Indeed, in the same study, patients with MN who achieved a complete remission after 6 months of immunosuppressive treatment had a differential expression of several IGHV genes, although the specific details on the immunosuppressive treatments used were not reported [102].

Analysis of the length of the third complementarity-determining region of the heavy chain (CDR-H3) showed that the length of the CDR-H3 loop for different immunoglobulin isotypes was significantly increased in patients with MN as compared to healthy controls, except for IgE. The CDR-H3 loop of the IgM and IgD isotypes were also significantly more hydrophobic in MN patients [102]. In healthy subjects, selection against long and hydrophobic CDR-H3 segments typically occurs at multiple stages of B cell development, such as through deletion or receptor editing in the naïve B cell compartment, or through negative selection and apoptosis in the mature compartment. Increased CDR-H3 length and hydrophobicity have been associated with autoreactivity and polyreactivity, through interference with IgH and IgL pairing [105,106]. Paradoxically, patients with SLE have typically been associated with short CDR-H3 segments, although it has been hypothesized that this may be due to a disproportionately large population of circulating plasmablasts in SLE [107]. It may therefore be helpful to investigate specific B cell subsets in MN patients to determine the site at which immune tolerance against self-reactive, long or hydrophobic IgH segments is breached.

Indirect evidence points to the loss of peripheral tolerance in secondary lymphoid organs (SLOs) in MN. Analysis of IGHA, IGHD, IGHG, and IGHM transcripts in PBMCs in MN showed an augmented rate of SHM, particularly for the IgG isotype [102]. Furthermore, inhibition of the CD40/CD40L pathway prevented development of murine MN, and blockade of the same pathway using a CD40 DNA vaccine targeted to dendritic cells protected predisposed rats from developing Heymann nephritis [108,109]. These findings demonstrate that failure of tolerance mechanisms to abrogate autoreactive T cell help via the CD40/CD40L co-stimulatory pathway is pivotal to the immunopathogenesis of MN.

Meanwhile, there is conflicting evidence regarding the expression of Bregs, which usually regulate and suppress memory B cell development arising from T_FH_-B cell interactions, in the context of MN. Whereas some studies have suggested Bregs were decreased in patients with MN, and the population rose with successful treatment of MN, others showed higher concentrations of Bregs compared to healthy controls and patients with non-immune causes of chronic kidney disease, though this may be a compensatory response rather than disease-causing per se [99,110,111,112]. It is crucial to note that Bregs are composed of various B cell subpopulations, including CD19^+^CD5^+^CD1d^hi^ cells that secrete IL-10, CD19^+^CD5^+^GzmB^+^ cells that secrete granzyme B, as well as other classes of B cells, and that the limited studies looking at Bregs in MN have focused primarily on IL-10-secreting Bregs. Furthermore, these studies mostly evaluated quantitative differences in Bregs, rather than the functionality of the isolated Bregs; importantly, in SLE and other autoimmune conditions, Bregs tended to produce less IL-10 and lacked suppressive capacity [113]. Thus, standardized classification of B cell subsets including Breg populations, analysis of circulating cytokine levels and functional assessment of Bregs are needed to better delineate the role of Bregs in MN and characterize their responses to immunomodulatory treatments.

Key B cell-related cytokines are upregulated in MN. Consistent with observations in other antibody-mediated autoimmune diseases, serum BAFF and APRIL levels were elevated in MN, although this was mostly observed in the group with detectable anti-PLA_2_R antibodies [114,115]. The different molecular profiles observed in different types of MN suggest that the mechanism of loss of immune tolerance may depend on the target antigen in question. Baseline serum baseline BAFF and APRIL levels were lower in patients who achieved a complete or partial response to standard immunosuppressive treatments as opposed to patients who had a limited or no response, and a marked reduction in the circulating levels of these cytokines was associated with favourable clinical outcomes [115]. In a separate study, high serum baseline APRIL was also associated with less complete response, while high serum baseline BAFF was associated with relapse [114]. BAFF also provides signals for SHM in SLOs, although no studies have yet explored the relation between serum BAFF levels and the level of SHM in B cells of MN patients [89].

## 5. Infiltrating B Cell Subsets and Intrarenal Tertiary Lymphoid Structures in Membranous Nephropathy

Early studies have highlighted that interstitial infiltrates are often present in MN and may contribute to development of tubulointerstitial fibrosis, attrition of nephrons and development of progressive CKD and ESKD [116]. Although these infiltrates were previously thought to be predominantly made up by macrophages, monocytes and T cells, recent studies have demonstrated prominent cellular infiltrates positive for the B cell marker CD20 in human MN, which were absent in patients with minimal change disease and healthy controls [117]. In association with such B cell infiltrates, other studies have shown that organized tertiary lymphoid structures (TLSs) may be found in human kidney biopsies with MN; the histological and immunohistochemical findings were corroborated by confirmation of expression of cytokines such as CXCL13 and lymphotoxin B (LTB), which are crucial mediators for the formation of TLSs [118,119]. TLSs are structures that develop in chronically inflamed organs as ectopic lymphoid aggregates resembling lymph nodes, with a variable degree of sophistication in its architecture, ranging from mere cellular aggregates of B and T cells to an SLO-like organization with B cell follicles, germinal centres or even lymphatic vessels. Such structures displace normal organ parenchyma and perpetuate organ-specific autoimmunity and have been described in a range of autoimmune disorders [120]. These include synovial tissues in rheumatoid arthritis, in salivary glands in Sjögren’s syndrome, in the thyroid in Hashimoto’s thyroidits, and in the kidneys in lupus nephritis, IgA nephropathy or kidney allograft rejection, among other end-organs [119,121,122]. In the aforementioned examples of autoimmune disease, TLSs promote local autoimmunity through a number of pathways. These include facilitation of local antigen presentation by B cells and other antigen-presenting cells, and provision of critical survival signals for B cells and long-lived plasma cells, including BAFF and IL-7 [120]. This results in B cell maturation and survival with consequent production of self-reactive antibodies directed against locally expressed antigens.

Although it is still conceivable that the intrarenal B cells migrated to the kidney from SLOs, the evidence hints at an in-situ tolerance break, with local presentation of podocyte antigens and B cell affinity maturation in intrarenal TLSs. This is supported by a clonal analysis of the B cell repertoire in immune-mediated glomerular diseases, which suggests the presence of an antigen-specific process of SHM and CSR that occurs locally in the renal parenchyma. Intrarenal oligoclonal B cells were found in glomerular diseases characterized by podocyte injury, including focal segmental glomerulosclerosis and MN, in the absence of B cell oligoclonality in the peripheral blood [123]. Furthermore, T cell interstitial infiltrates in MN usually show an elevated CD4^+^/CD8^+^ ratio [116]; and in human and experimental models of MN, the Th2 polarization of CD4^+^ T cells stimulates both peripheral and intrarenal B cells to produce IgG4, the culprit Ig isotype in MN [124]. A small study of MN patients has also shown that BAFF and APRIL are elevated not just in the sera but also in the renal tissue [114]. Taken together, these findings suggest that even if kidney-specific autoimmunity is not initiated by TLSs, it is exacerbated and perpetuated by their existence, leading to chronic damage and increased propensity for disease relapse.

While there was no correlation between such infiltrates and proteinuria, patients with an abundance of B cell infiltrates on the kidney biopsy were more likely to have a lower estimated glomerular filtration rate. B cell depletion with rituximab in such cases resulted in significantly better kidney function in the long term, underscoring the role of interstitial inflammation in chronic kidney damage and fibrosis in glomerular diseases [118]. However, studies have not specifically analyzed the subsets of B cells present in these interstitial infiltrates in detail. In the context of renal transplantation, pre-emptive induction treatment with rituximab paradoxically increased the risk of cellular rejection, presumably because of non-specific removal of all B cell infiltrates including Bregs [125]. The contribution of Bregs to the interstitial infiltrates observed in MN would therefore be of interest and clinical relevance.

## 6. B Cells and the Treatment of Membranous Nephropathy

The conceptualization of MN as a B cell disorder has revolutionized its treatment. Conventionally, MN that failed to resolve spontaneously with conservative medical management was treated with either a cyclical combination of corticosteroids and cyclophosphamide (such as the modified Ponticelli regimen), or calcineurin inhibitors such as cyclosporin A and tacrolimus, with or without concomitant corticosteroids [126,127]. Although the former option was highly potent and demonstrated favourable long-term outcomes, the use of cyclophosphamide was burdened with significant adverse effects including infertility, infections, urological toxicity and secondary malignancies [128]. Meanwhile, the use of calcineurin inhibitors was associated with a high rate of relapse upon cessation of therapy, necessitating long-term use but with the attendant risks of calcineurin inhibitor side effects, which include chronic nephrotoxicity, worsening of hypertension, diabetes and dyslipidaemia [129]. The use of B cell depletion in MN thus holds promise as a relatively less toxic treatment option, premised on a more sophisticated understanding of the immunopathogenesis of MN, and recently promulgated guidelines support the use of rituximab as first-line treatment for moderate-to-high-risk MN [126,127].

The efficacy of various treatments for MN may be, at least in part, related to their impacts on the B cell signature of the disease. Although there is scant data on the impact of cyclophosphamide on B cell markers in the specific context of MN, the extensive experience with cyclophosphamide in other diseases suggests cyclophosphamide effectively decreases both the B cell and T cell compartments, especially the former [130]. Through its action on DNA synthesis, cyclophosphamide inhibits various stages of B cell proliferation and differentiation, and also exhibits substantial activity against plasmablasts and short-lived plasma cells, thereby significantly reducing the generation of new antibody-secreting cells [131,132,133]. In the context of MN, serum levels of BAFF and APRIL were both reduced after treatment with the modified Ponticelli regimen. Conversely, calcineurin inhibitors are T cell-targeted therapies that mediate indirect effects on the B cell population via modulating T cell help to B cells. While cyclosporin A seems to be primarily effective against B cells at earlier stages of development prior to their activation, tacrolimus inhibits both the activation and proliferation of human B lymphocytes in vitro, and reduces immunoglobulin secretion after antigen stimulation [134,135]. However, neither cyclosporin A nor tacrolimus affects B cell differentiation [136,137]. Importantly, in studies of MN patients, both cyclophosphamide- and calcineurin inhibitor-based treatment regimens are clinically useful in MN, and can effectively reduce anti-PLA_2_R antibody titres in PLA_2_R-associated MN, although greater, earlier and more sustained decreases were typically seen with cyclophosphamide-based regimens [138]. Patients on calcineurin inhibitors do not tend to go into durable remissions and have a high rate of immunological and clinical relapse after cessation of treatment.

B cell depletion, most commonly with the chimeric anti-CD20 monoclonal antibody rituximab, has been associated with high rates of immunological and clinical response in the context of MN. Analysis of B cell and T cell subpopulations in rituximab-treated patients with autoimmune kidney disease shows that rituximab depletes B cells effectively in the peripheral blood, particularly CD40^+^ memory B cells, switched memory B cells and plasmablasts [139]. Typically, in responders, B cell depletion is followed days to weeks later with decreases in anti-PLA_2_R antibody titres in seropositive patients, and this is later followed by a clinical response [140]. B cell depletion is additionally associated with a decrease in pro-inflammatory cytokines, particularly tumor necrosis factor-α (TNF-α), though it remains speculative whether these effects may be attributable to B cell depletion alone. In MN, there are so far no reliable B cell signatures that would predict a clinical and immunological response to rituximab, but responders tend to show a lower percentage of Tregs at baseline and an increased percentage shortly after treatment [98,139]. This suggests that part of the effect of rituximab could be mediated by an increase in the Treg fraction, which affects mechanisms of peripheral B and T cell tolerance. This effect may be related to reduced antigen presentation by memory B cells to T cells, and attenuation of B cell-related cytokine production, including interferon-γ.

Various treatment regimens for rituximab have been tried, and both the dosing regimens of rituximab 375 mg/m^2^ weekly for 4 weeks and rituximab 1 g every 2 weeks for 2 doses have been shown to be effective in achieving complete B cell depletion, and immunological and clinical responses, in MN [56,57]. Smaller studies suggest that titrating rituximab to circulating B cell levels may optimize total rituximab dose, reduce treatment cost and possibly minimize side effects [141]; although a recent retrospective analysis of MN patients treated with higher doses compared to lower doses of rituximab suggested that the full, high-dose protocol was associated with higher remission rates, a shorter median time to achieving remission, and more profound B cell depletion [142,143]. This may be especially important for patients with high circulating anti-PLA_2_R antibody titres before treatment. Further investigation of different dosing schedules of rituximab in MN are likely indicated: compared to other autoimmune diseases for which rituximab is used, including ANCA-associated vasculitis (AAV), rheumatoid arthritis and myasthenia gravis, the half-life is shorter and serum levels of rituximab are lower in MN patients, likely because of significant urinary wasting of rituximab in the context of nephrotic-range proteinuria due to MN; these decreases have been associated with earlier peripheral B cell reconstitution, higher anti-PLA_2_R titres and higher degrees of proteinuria at various time points [144,145,146].

The tempo of B cell repopulation following transient depletion differs among various autoimmune diseases, and may affect the timing of disease relapse and the choice of maintenance immunosuppression. Among patients with AAV, only 10% of patients with granulomatosis with polyangiitis (GPA) or microscopic polyangiitis (MPA) had any evidence of B cell repopulation at 1 year after a standard rituximab-based induction, and no patients with eosinophilic granulomatosis with polyangiitis (EGPA) had detectable B cells in the peripheral blood, whereas B cell repopulation had commenced in over 90% of patients with rheumatoid arthritis at 1 year [147]. This may suggest either latent B cell defects associated with the underlying disease, or the effects of other concomitant immunosuppressive treatments, such as azathioprine or mycophenolate mofetil in the context of AAV. B cell levels over time were not consistently reported in all studies of rituximab in MN, but small studies suggest that B cell repopulation typically occurs at around 9 to 12 months for patients receiving standard, high-dose rituximab for induction [142]. Although total B cell count, i.e., the CD19^+^ cell count, is often used in clinical practice to gauge the response to anti-CD20 treatments, it is not well-established as to whether sustained total peripheral B cell depletion alone is a suitable treatment target in MN to prevent relapse. For cases of MN that have entered into remission, small studies investigating B cell subsets in MN have suggested that anti-PLA_2_R antibody-secreting plasmablasts and memory B cells may reappear before naïve B cells, leading to relapse [148]. On the contrary, sustained immunological and clinical remission may be seen even in the context of B cell repopulation, and re-treatment to ensure continued B cell depletion might well be unnecessary or even counterproductive [53]. This may be related to changes in the B cell repertoire following transient depletion, the effects of which may not be captured by simply monitoring the total CD19^+^ cell count. Studies in other primary glomerulonephritides such as childhood idiopathic nephrotic syndrome have demonstrated prolonged impairment of immunological memory following treatment with anti-CD20 therapies, with sustained reductions in total memory B cells and switched memory B cells [149]. In some cases, these could be associated with significant hypogammaglobulinaemia. Similar B cell repopulation kinetics have been observed in rheumatoid arthritis, with a slow and delayed repopulation in the memory B cell subset, resulting in persistently suppressed memory B cell levels for over 2 years after the last dose of rituximab [150]. Importantly, the use of rituximab in conjunction with corticosteroids in neuromyelitis optica spectrum disorders and in immune thrombocytopenia have additionally been associated with an increased proportion of Bregs upon repopulation, suggesting that B cell depletion with anti-CD20 monoclonal antibodies may actually restore immune tolerance in certain autoimmune conditions [151,152]. Associated B cell biomarkers and cytokines may also be relevant: in autoimmune kidney disease including MN, treatment with rituximab and subsequent B cell depletion was paralleled by a compensatory increase in APRIL levels [139]; there is no data specific to the MN population to suggest whether such an increase may be associated with a higher risk of relapse or treatment failure. Meanwhile, rheumatoid arthritis and SLE, rituximab treatment is typically associated with compensatory elevations in BAFF and this may herald relapse in disease activity [153,154,155,156]. Whether these B cell biomarkers carry the same prognostic implications in MN remains speculative.

Despite the promise of rituximab treatment, there are still significant unmet treatment needs. The remission rate achieved by rituximab in MN remains at only 60–70%, suggesting that some components of the immune pathways leading to MN are not adequately addressed and that this may lead to resistance to rituximab [53,144]. Given the aforementioned increased BAFF levels in MN patients, and the knowledge that BAFF levels are elevated in other autoimmune diseases following rituximab treatment, anti-BAFF treatment such as with belimumab (a human IgG1-λ monoclonal antibody against BAFF approved for the treatment of SLE and lupus nephritis) have been trialed in MN. Phase 2 studies have encouragingly shown that belimumab can effectively reduce proteinuria and anti-PLA_2_R antibody levels in MN [60]. However, in class V lupus nephritis, a form of secondary MN, belimumab did not improve rates of renal response as compared to standard of care [157]. To settle this question, a randomized controlled study investigating sequential treatment of MN with rituximab followed by belimumab is currently under way (NCT03949855). More information regarding the molecular and cellular signature in responders would be helpful, as indiscriminate targeting of BAFF may also result in depletion of Bregs and paradoxically worsen autoimmunity [158]. There is comparatively little experience with use of anti-APRIL therapies in MN or other autoimmune diseases; a phase 2/3 study of atacicept, a human recombinant fusion protein of TACI and the Fc portion of IgG1, in combination with mycophenolate mofetil in SLE was discontinued prematurely due to hypogammaglobulinaemia and opportunistic infections [159].

Meanwhile, newer, humanized anti-CD20 monoclonal antibodies, such as obinutuzumab and ofatumumab, have been investigated in MN, especially in rituximab-resistant disease [58,59,61,160,161]. Type II anti-CD20 monoclonal antibodies generally confer more potent B cell depletion compared to rituximab, a type I antibody, as type I antibodies lead to antigenic modulation of CD20 with resultant internalization of the CD20/antibody complex in B cells, reduced antibody-dependent phagocytosis and consumption of the monoclonal antibody. Due to its chimeric structure, administration of rituximab has also been associated with the development of anti-drug antibodies [144]. Obinutuzumab binds to a different epitope of CD20 to avoid internalization of the CD20/antibody complex, and is also glycoengineered to generate a higher level of antibody-dependent cell cytotoxicity, with reduced dependence on complement-dependent cytotoxicity. It is worthwhile to note that persistence of TLSs in tissues as well as in lymph nodes despite peripheral B cell depletion by rituximab is well documented in a number of autoimmune diseases, including rheumatoid arthritis and Sjögren’s syndrome [162]. The finding of B cell-containing TLSs in MN suggests that deeper B cell depletion may be beneficial for refractory disease. Although it remains speculative whether type II anti-CD20 monoclonal antibodies are truly more effective in eradicating TLSs, depletion of B cells in lymph nodes in addition to the peripheral blood was demonstrated in a pilot study of obinutuzumab for desensitization of highly sensitized kidney transplant candidates [163]. Theoretically, even more extensive B cell depletion can be achieved with chimeric antigen receptor T cells (CAR-T), a treatment currently approved for refractory B cell lymphomas and leukaemias [164]. However, CAR-T is associated with high treatment costs and significant toxicities, and hence its use in a less life-threatening and relatively indolent condition like MN appears unjustified at present.

Finally, certain antibody-secreting cells such as plasmablasts or plasma cells do not express the CD20 marker, and the effects of anti-CD20 treatment on these cell types are often indirect and incomplete [165]. Indeed, in the largest published study comparing a cyclophosphamide-based treatment regimen with a rituximab-containing regimen, both immunological response including the tempo and magnitude of decline in anti-PLA_2_R antibody levels, as well as clinical response, were superior in the cyclophosphamide arm, although some studies show that rituximab may be safer and better tolerated [55,166]. Rituximab monotherapy has also been shown to have a higher rate of failure in patients with a high baseline titre of anti-PLA_2_R antibody [167]. These differences may possibly be due to the fact that cyclophosphamide is more effective than rituximab in targeting such antibody-secreting cells. Efforts to overcome these hurdles include use of combination rituximab and cyclophosphamide for high-risk cases [168]; or the use of anti-plasma cell therapies such as the proteasome inhibitor bortezomib, with or without corticosteroids or other immunosuppressive treatments [169,170]. From an immunological perspective, the latter option would be able to eradicate long-lived plasma cells in the bone marrow, which are often even resistant to alkylating agents such as cyclophosphamide. An ongoing phase 2 study is exploring the safety of MOR202, an anti-CD38 monoclonal antibody targeted at plasma cells, in PLA_2_R-associated MN (NCT04145440). The effects of these novel biologic therapies are summarized in Figure 2. Larger studies are evidently needed to elucidate the role of these multimodal rescue therapies. Ultimately, given that none of these therapies are truly specific for MN, there is often a trade-off between greater treatment potency and off-target adverse effects, some of which can be potentially severe or even fatal.

## 7. Conclusions

MN is a kidney-specific autoimmune disease involving a loss of central and peripheral tolerance to autoreactive B cells. In recent years, a more sophisticated understanding of the underlying immunopathogenesis of MN, especially the discovery of disease-associated autoantigens and autoantibodies, has led to a shift in the treatment paradigm from non-specific immunosuppression to B cell-targeted therapies. Further research is needed to better delineate the mechanisms by which B cell tolerance is breached in MN, to guide the development of more specific and effective treatments. In this regard, elucidation of the B cell signatures of MN patients prior to enrolment in clinical trials and assessment of changes in B lymphocyte and plasma cell subpopulations with treatment may help identify suitable groups of patients for novel therapies. Finally, correlation of B cell-related cellular biomarkers and cytokines with histological and clinical parameters may provide a non-invasive tool for prognostication and treatment selection in MN, leading to improved outcomes in this patient population.

## Figures and Tables

**Figure 1 ijms-22-13560-f001:**
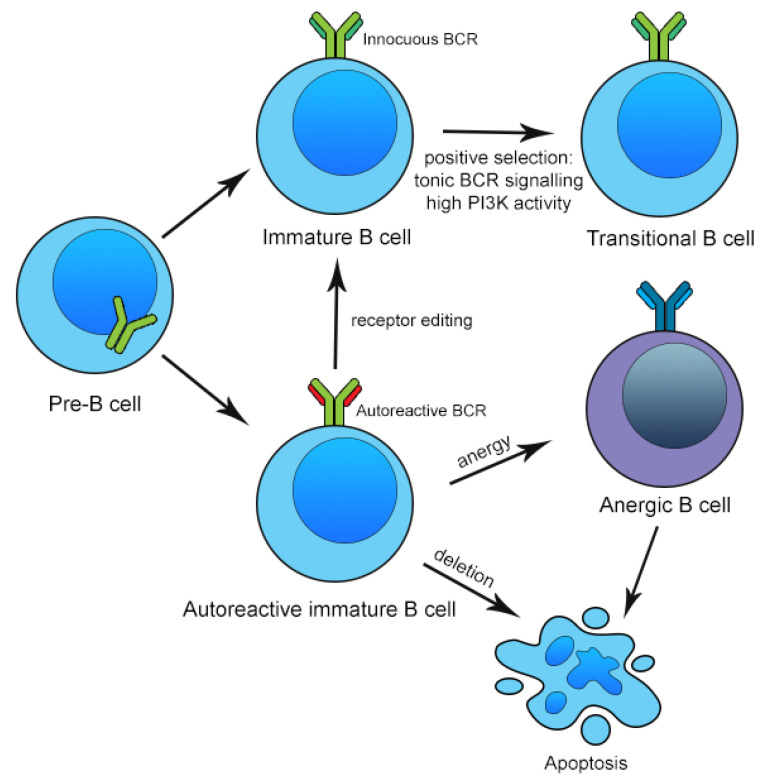
Mechanisms of central B cell tolerance in the bone marrow.

**Figure 2 ijms-22-13560-f002:**
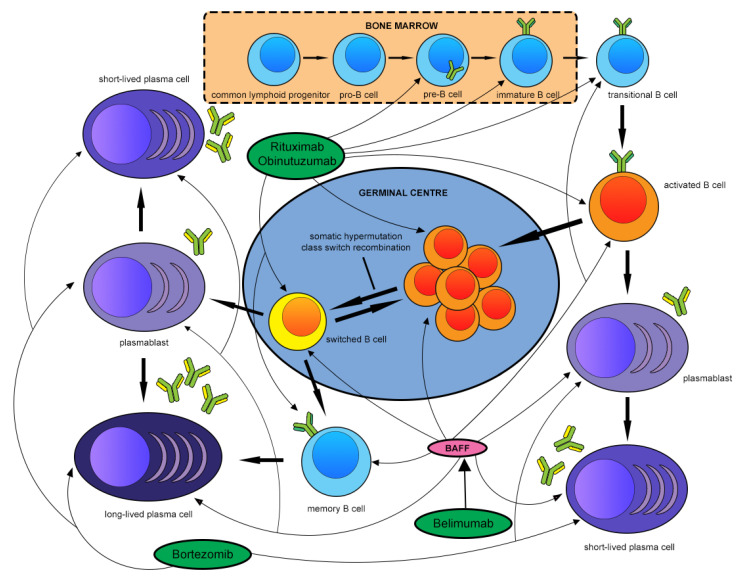
Effect of biologics targeted against B cells and plasma cells on various stages of B cell development in membranous nephropathy.

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
