# Peer review of "B Cells in Primary Membranous Nephropathy: Escape from Immune Tolerance and Implications for Patient Management"

_ijms, 2021, doi:10.3390/ijms222413560_

Round 1

Reviewer 1 Report

This is an excellent paper. I pay tribute to the authors. There are five comments below.

1. Belimumab appears to be less effective in class V (MN) than class III/IV in lupus nephritis (see literature below). This information may be considered in your paper.

Rovin BH, Furie R, Teng YKO, Contreras G, Malvar A, Yu X, Ji B, Green Y, Gonzalez-Rivera T, Bass D, Gilbride J, Tang CH, Roth DA. A secondary analysis of the Belimumab International Study in Lupus Nephritis trial examined effects of belimumab on kidney outcomes and preservation of kidney function in patients with lupus nephritis. Kidney Int. 2021 Sep 22:S0085-2538(21)00862-0.

2. It is better to make the letters in Figure 1 larger.

3. B cell and B-cell or T cell and T-cell are mixed. It is better to unify.

4. P. 4, L. 135:
It may be better to write follicular helper T (TFH) cells instead of follicular T helper cells (TFH cells).

5. P. 6, L. 229:
Study is in italic font. Please correct it to a regular font. 

Author Response

1) Thank you for the suggested reference. It has been added to the article, in the discussion on the potential role of belimumab in membranous nephropathy.

2) The image has been updated for clarity.

3 - 5) The stylistic changes have been incorporated.

Reviewer 2 Report

Excellent overview of the meaning of B Cells in Primary Membranous Nephropathy, but requiring more detailed disclosure of the following issues:

  1. Which B cell-related cellular biomarkers we can use to diagnose and predict the course of the disease
  2. The role of regulatory B cells should be described in more detail

Author Response

Thank you for the comments. Our responses are detailed below:

1) While the B cell repertoire plays an instrumental role in the pathogenesis of MN, there is no distinct cellular marker or B cell subtype that shows good diagnostic value yet. In clinical practice, the total B cell count (i.e. CD19+ B cells) rather than individual B cell subset is often measured to gauge the degree of B cell depletion in patients receiving anti-CD20 treatments. We have added several lines to the relevant paragraphs in the article to reinforce this point regarding the changes in B cell populations following treatment, especially anti-CD20 monoclonal antibodies.

2) Thank you for the suggestion. We have added further discussion on the potential role of regulatory B cells in MN, with reference to their purported role in other autoimmune diseases, and highlighted key areas in which further research is required.